



# 1 Validation of a new cavity ring-down spectrometer

# 2 for measuring tropospheric gaseous hydrogen

# 3 chloride

*Teles C. Furlani[1], Patrick R. Veres[2], Kathryn E.R. Dawe[3, a], J. Andrew Neuman[2, 4], Steven S.*
*Brown[2,5], Trevor C. VandenBoer[1], Cora J. Young[1]*
[1] Department of Chemistry, York University, Toronto, ON, Canada
[2] NOAA Chemical Sciences Laboratory, Boulder, CO, USA
[3] Department of Chemistry, Memorial University of Newfoundland, St. John's, NL, Canada
[4] Cooperative Institute for Research in Environmental Sciences, University of Colorado, Boulder, CO, USA
[5] Department of Chemistry, University of Colorado, Boulder, CO, USA
[a] Now at SEM Ltd., St. John's, NL, Canada
**Abstract**
Reliable, sensitive, and widely available hydrogen chloride (HCl) measurements are important for
understanding oxidation in many regions of the troposphere. We configured a commercial HCl
cavity ring-down spectrometer (CRDS) for sampling HCl in the ambient atmosphere and
developed calibration and validation techniques to characterize the measurement uncertainties.
The CRDS makes fast, sensitive, and robust measurements of HCl in a high finesse optical cavity
coupled to a laser centered at 5739 cm$^{-1}$. The accuracy was determined to reside between 5–10%,
calculated from laboratory calibrations and an ambient air intercomparison with annular denuders.
The precision and limit of detection (3$\sigma$) in the 0.5 Hz measurement were below 6 pptv and 18





pptv, respectively for a 30 second integration interval in zero air. The response time of this method
is primarily characterized by fitting decay curves to a double exponential equation and is impacted
by inlet adsorption/desorption, with these surface effects increasing with RH and decreasing with
decreasing HCl mixing ratios. The response time for the tested inlet was 2–6 minutes under the
most and least optimal conditions, respectively. An intercomparison with the EPA compendium
method for quantification of acidic atmospheric gases showed good agreement, yielding a linear
relationship statistically equivalent to unity (slope of $0.97 \pm 0.15$). The CRDS from this study can
detect HCl at atmospherically relevant mixing ratios, often performing comparable or better in
sensitivity, selectivity, and response-time from previously reported HCl detection methods.

## 1. Introduction

Halogenated compounds that participate in catalytic cycles in the atmosphere have major
impacts on atmospheric chemistry. Chlorine-containing species have long been known to
catalytically destroy stratospheric ozone (Solomon, 1999) and can have similar impacts on
tropospheric ozone in polar regions (Simpson et al., 2007, 2015). In particular, early morning
oxidation in the troposphere can be influenced heavily by chlorine atoms released by photolabile
chlorine species (Osthoff et al., 2008; Thornton et al., 2010; Young et al., 2012, 2014). It is
estimated that reactions involving chlorine atoms account for 14–27% of global tropospheric
oxidation of abundant volatile organic compounds (VOCs) (Sherwen et al., 2016).
The role of chlorine chemistry in the troposphere remains uncertain in part due to a lack of
a complete understanding of the contribution of chlorine reservoir species to the tropospheric
chlorine inventory (Osthoff et al., 2008; Young et al., 2014). Sources of inorganic chlorine to the
troposphere are important because many of them are photochemically active (e.g. $ClNO_2$). A near-



complete budget of inorganic tropospheric chlorine from aircraft transects of polluted North
American continental outflow during the WINTER campaign demonstrated that hydrogen chloride
(HCl) makes up 48–62 % of total inorganic chlorine, and approximately 98% of total gaseous
inorganic chlorine (Haskins et al., 2018). Troposphere HCl levels are typically between 10 and
1000 parts per trillion by volume (pptv) (e.g. Crisp et al. (2014); Haskins et al. (2018); and Young
et al. (2013)).
HCl is directly emitted to the atmosphere predominantly from volcanic activity, biomass
burning, and industrial sources (Butz et al., 2017; Crisp et al., 2014; Keene et al., 1999). HCl is
also produced through heterogeneous acid displacement reactions of strong acids, such as nitric
acid, with particulate chloride (pCl⁻) (Bondy et al., 2017; Clegg and Brimblecombe, 1985; Gard et
al., 1998; Valach, 1967; Wang et al., 2019):
$HA_{(g, aq)} + MCl_{(aq, s)} \rightarrow MA_{(aq, s)} + HCl_{(g)}$ (R1)
where M represents a cation in a chloride salt (often sodium). Elevated levels of chlorine atoms
may also be present in both indoor and outdoor environments due to the emission of photolabile
reactive chlorine compounds (Dawe et al., 2019; Mattila et al., 2020; Osthoff et al., 2008; Young
et al., 2014, 2019).
$XCl_{(g)} + h\upsilon \rightarrow Cl\cdot_{(g)} + X\cdot_{(g)}$ (R2)
Secondary production of HCl predominantly occurs *via* the reaction of chlorine atoms with
methane or VOCs by hydrogen abstraction:
$Cl\cdot_{(g)} + CH_{4(g)} \rightarrow HCl_{(g)} + \cdot CH_{3(g)}$ (R3)
$Cl\cdot_{(g)} + RH_{(g)} \rightarrow HCl_{(g)} + R\cdot_{(g)}$ (R4)

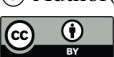



The loss of gas-phase HCl occurs predominantly through wet or dry deposition which are terminal
sinks for tropospheric chlorine (Wang et al., 2019), with minor loss by reaction with the hydroxyl
radical (OH) to re-form chlorine atoms:
$\qquad HCl_{(g)} + OH\cdot_{(g)} \rightarrow Cl\cdot_{(g)} + H_2O_{(g)}$ $\qquad\qquad$ (R5)
The balance between loss and formation of chlorine atoms from HCl is highly dependent on factors
such as the presence of $pCl^-$, $NO_x$ ($NO_x=NO+NO_2$), and HCl deposition rate (Finlayson-Pitts et
al., 1989; Roberts et al., 2008).
$\qquad$ Measuring HCl in the gas phase is challenging as it readily adsorbs to surfaces. Methods
for atmospheric HCl detection must be sensitive, robust, and selective and address HCl interactions
with instrument surfaces. Mass spectrometry based measurement techniques have been developed
for the detection of HCl in both the stratosphere and troposphere (e.g. Huey et al. (1996) and Marcy
et al. (2004)). Other methods include scrubbing ambient air using an annular denuder and/or a
tandem mist chambers to collect HCl, followed by offline analysis such as ion chromatography
(IC) (Keene et al., 2007, 2009; United States Environmental Protection Agency, 1999; Young et
al., 2013). Online detection methods such as chemical ionization time of flight mass spectrometry
(CI-ToF-MS) (Crisp et al., 2014), negative ion proton transfer chemical ionization mass
spectrometry (NI-PT-CIMS) (Veres et al., 2008), and negative mode atmospheric pressure
chemical ionization coupled to triple quadrupole mass spectrometry (APCI-MS-MS) (Karellas et
al., 2003) have been shown to be reliable and sensitive methods for HCl detection. Limitations to
these existing HCl measurement techniques include some or all of the following; detection limits
that are not suitable for the low level of HCl in the troposphere, slow time response, lack of
portability, and calibration challenges.





Measurements of HCl are typically calibrated using HCl from permeation devices and or
standards in compressed gas cylinders. Method validation for HCl measurements are rare, but can
reduce these uncertainties (Hagen et al., 2014). In this paper we demonstrate the versatility and
validation of a new commercial cavity ring-down spectrometer (CRDS) for in-situ atmospheric
gas phase HCl measurements. We compare this CRDS with existing HCl measurement techniques
through lab and field intercomparisons. Finally, we describe and characterize surface effects and
recommend inlet configurations for best practices when conducting ambient sampling.
**2. Materials and experimental methods**
**2.1 Chemicals**
Reagent grade hydrochloric acid (HCl, 12 M) was used in permeation device construction
(see Section 2.3). Potassium hydroxide (KOH) pellets were used to create scrubbing solution for
permeation device gas collection. Commercially available reagents were from Sigma-Aldrich
(Oakville, Ontario, Canada) and Ultra zero air (grade 5.0) gases were from Praxair (Toronto,
Ontario, Canada). Experiments used deionized water generated by a Barnstead Infinity Ultrapure
Water System (Thermo Fisher Scientific, Waltham, Massachusetts, USA; $18.2\ \mathrm{M\Omega\ cm}^{-1}$). Annular
denuder coating solution was prepared with reagent grade (>95.5%) sodium carbonate (Sigma-
Aldrich, St. Louis, Missouri, USA), reagent grade glycerol (Sigma-Aldrich, St. Louis, Missouri,
USA), HPLC grade methanol (Fisher Chemicals, Ottawa, Ontario, Canada), and $18.2\ \mathrm{M\Omega\ cm}^{-1}$
deionized water. Eluent for annular denuder IC analysis was prepared from sodium hydroxide
solution (NaOH, 50% w/w, Thermo Fisher Scientific, Sunnyvale, California, USA). IC calibration
standards were prepared through serial dilution of a mixed anion standard concentrate (Thermo
Fisher Scientific, Dionex Seven-Anion II, P/N: 057590). Nitrogen (grade 4.8).
**2.2. Cavity ring-down spectrometer (CRDS)**



The Picarro G2108 Hydrogen Chloride Gas Analyzer system was used for all analyses
(Dawe et al., 2019, www.picarro.com). The basic operating principles of this CRDS are similar to
analogous Picarro greenhouse gas instruments that have been described in detail by Crosson et. al
(2008). The CRDS consists of a tunable laser, a wavelength monitor, and a heated optical cavity
(80 °C). All the components of this analyzer and internal stainless-steel fittings are contained
within a heat-regulated metal case maintained at 45 °C. The laser radiation (1742 nm, 5739 cm$^{-1}$)
is directed by a fiber optic cable to the wavelength monitor and optical cavity. The first overtone
(2-0 absorption band) of HCl is easily discernable from other absorbing species (e.g. $H_2O$, $CH_4$),
has a relatively high intensity, and is accessible to near-infrared (IR) diode laser light sources. The
optical cavity is fitted with three highly reflective dielectric-coated fused silica mirrors (R >
99.995%, ring down time constant of 53 μsec, equivalent to a path length of 16 km) oriented in an
acute triangular arrangement supported by an invar housing. The reflectivity of the mirrors is
measured from the laser signal loss in an analyte-free optical cavity under inert gas flow. The
CRDS flow rate is 2 L min$^{-1}$ and the cavity is held at a reduced pressure of 18.70 ± 0.02 kPa (140
Torr) thermostated to 80.000 ± 0.005°C. One mirror is mounted on a piezoelectric actuator to
achieve optical resonance between the laser frequency and the longitudinal modes of the cavity.
The laser is shut off rapidly (< 1 μsec) once resonance is achieved. A photodetector monitors the
decay of the photons exiting the cavity through another mirror. Custom electronics digitize the
signal for fitting of an exponential decay; the time constant of the decay, τ, is converted to
absorbance, α, using the expression
$$\alpha = 1 \, / \, c\tau - 1 \, / \, c\tau_0 \quad \text{E1}$$

where c is the speed of light. The instrument measures 30 specific frequencies within ~1 cm$^{-1}$
centered at 5739 cm$^{-1}$ to fit the absorption spectra of trace species in this region (see Figure S1).



HCl, $H_2O$, and $CH_4$ mixing ratios are reported every 2 seconds, though the true time response of
the measurement method is limited by surface effects (see Section 3.4). Gaseous inorganic chlorine
reservoir species (e.g. $ClNO_2$) cannot thermally dissociated under the cavity (80 °C) conditions
(Thaler et al., 2011). The instrument zero measurement drift is reduced by a high precision
distributed feedback laser centered at 5739.2625 $cm^{-1}$ coupled with a custom-designed wavelength
monitor to determine the frequency axis of each spectrogram. To mitigate particulate matter optical
extinction and surface deposition on the high reflectivity mirrors, two high efficiency particulate
air (HEPA) filters are placed upstream of the cavity, contained within the 45 °C heat-regulated
compartment.

**2.3 In-house HCl permeation device validation**

143        The in-house assembly of HCl permeation devices (PDs) is described in detail in Lao et al.

(2020). Briefly, 200 µL of 12 M aqueous HCl solution was pipetted into a 7.62 cm perfluoroalkoxy
(PFA) tube (3 mm i.d. with 1 mm thickness) plugged at both ends with porous
polytetrafluoroethylene (PTFE) (1 cm length by 3.17 mm o.d.). The polymers allow a consistent
mass of HCl to permeate at a given temperature and pressure. An aluminum block that was
temperature-controlled using a cartridge heater (Omega$^{TM}$; CIR-2081/120V, Saint-Eustache, QC,
Canada) housed the PD and was regulated to 60.0 ± 0.1 °C by a process controller. Dry $N_2$ flowed
through a PFA tube (1.27 cm o.d.) in the block, containing the PD. Stable flows of 49 ± 2 standard
cubic centimeters per minute (sccm) through the oven were maintained with a 50 µm diameter
critical orifice (Lenox laser, Glen Arm, Maryland, USA, 15 psi; SS-4-VCR-2-50). Flows were
measured using a DryCal Definer 220 (Mesa Labs, Lakewood, Colorado, USA). The mass
emission rate of HCl from the PD was quantified by scrubbing into a 25 mL glass impinger



containing 1 mM KOH over 24 h followed by analysis using IC with conductivity detection (CD).
Mass emission rates for the PD were determined as $140 \pm 18$ ng min$^{-1}$ (n=3, $\pm$ 1$\sigma$) at 60 °C.

**2.4 Laboratory intercomparison**

A laboratory intercomparison between the CRDS and offline-measured HCl scrubbed into
a basic solution of 100 mM KOH by delivering gaseous HCl from the permeation device to the
sampling systems. The 140 ng min$^{-1}$ of HCl in dry N$_2$ from the PD was mixed into a zero air
dilution flow of 2.1 to 8.0 L min$^{-1}$, to provide standard addition HCl calibrations that ranged from
12 to 45 ppbv.
The dilution flow was maintained using a 10 L min$^{-1}$ mass flow controller (GM50A, MKS
instruments, Andover, Massachusetts, USA). All inlet lines and fittings were kept at ambient
temperature (~25 °C) and were made of PFA unless stated otherwise. The inlet mixing line
between the PD emissions and the humidified dilution flow was 3.17 mm i.d. and 45 cm in length.
Residence times for HCl in the sampling line ranged from 0.02 to 0.08 seconds. To vary relative
humidity (RH), a controlled flow of zero air was directed into a glass impinger at room temperature
containing deionized water to yield a water-saturated air stream. The humidified flow was passed
through a 2 μm Teflon filter (TISCH scientific, North Bend, Ohio, USA) in a PFA holder to
prevent any aqueous droplets from entering the experimental lines. The RH was set by mixing
with dry zero air to generate 0, 20, 50, and 80 % RH values.

**2.5 Ambient intercomparison**

An ambient intercomparison was undertaken by measuring outdoor air with the CRDS in
parallel with sodium carbonate-coated annular denuders. A total of 11 denuder samples were
collected alongside continuous CRDS observations, each for approximately 24 hours between 4–
11 April 2019. The measurement site was the Air Quality Research Station, located on the roof of



the Petrie Science and Engineering Building at York University in Toronto, Ontario, Canada
(43.7738° N, 79.5071° W, 220 m above sea level). All indoor inlet lines and fittings were kept at
room temperature while the outdoor temperature ranged from -2 to 14 °C. All inlet lines and
fittings were made of PFA unless stated otherwise. A full schematic of the sampling apparatus
indicating the separation between the outdoor and indoor inlet positions is provided in Figure S4.
A 22 L min$^{-1}$ sampling flow was pulled through a URG Teflon Coated Aluminum Cyclone (URG
Corporation, Chapel Hill, North Carolina, USA) with a 2.5 µm cut-off for particulate matter. The
inlet lines were such that each sampling setup collected HCl at equal residence time to ensure
equivalent wall losses of HCl. The shared inlet line was 4.65 m in length and had an i.d. of 4.76
mm. The flow was split between the 1.5 m denuder sampling line (20 L min$^{-1}$) and the 0.15 m
CRDS sampling line (2 L min$^{-1}$), yielding a 0.375 sec residence time for both methods. The
denuder line flow was equally divided into two multichannel etched glass annular denuders (URG
Corporation, Chapel Hill, North Carolina, USA, 4 channel, 242 mm length, URG-2000-30x242-
4CSS) at 10 L min$^{-1}$. The denuders collected HCl in parallel to each other with flows controlled
using two separate 10 L min$^{-1}$ mass flow controllers (GM50A, MKS instruments). Denuders were
coated with a solution of 2% w/w sodium carbonate and 0.1% w/w glycerol in a solution of 1:1
methanol:water. A 15 mL aliquot of coating solution was dispensed into a denuder and two
polypropylene caps affixed. The sealed denuders were inverted and rotated for a few minutes to
ensure an even coating. The excess coating solution was decanted, and the denuder was dried for
15 min with 2 L min$^{-1}$ of zero air. After sampling, denuders were extracted with 2 aliquots of 5.00
mL deionized water, following the same sealing and inversion procedure, for a total extraction
volume of 10.00 mL. Extracts were collected into a 15 mL polypropylene tube for storage at 4 °C
until analysis. Instances of flagged instrument errors in the CRDS data during ambient



observations were removed as standard practice in quality control procedures (see Figure S2). The
loss of observational data during such periods corresponds to a negative bias. The CRDS data loss
during a given denuder sampling period was included in setting the overall measurement error
when intercomparing measurements.
**2.6 Ion chromatography analysis**

206        Samples collected into an impinger from the HCl PD were analyzed as the chloride anion

by IC-CD using an ICS-2100 (Thermo Fisher Scientific, Sunnyvale, California, USA) according
to the method described in Place et al. (2018). Annular denuder extracts were analyzed by IC-CD
using an ICS-6000 (Thermo Fisher Scientific, Sunnyvale, California, USA). Details of both
separation methods can be found in the SI. Chloride was quantified using external calibration with
a 5-point calibration curve. Two check standards, located at the high and low ends of the working
range, were used to evaluate the accuracy of quantification.
**3. Results and discussion**
**3.1 Suitability for atmospheric measurements**

215        The selectivity of this CRDS analyzer arises from monitoring a high-intensity spectral line

($5739.2625$ cm$^{-1}$). The absorption used by this instrument is suitable for HCl measurements in the
ambient atmosphere because abundant atmospheric gaseous species such as CO, $CO_2$, $NO_x$, and
$N_2O$ (Gordon et al., 2017; Kochanov et al., 2019) do not have major absorption features in the
same region. Absorption features of $H_2O$ and $CH_4$ in this spectral region are part of the fitting
parameters used to determine number densities of HCl, as described in Section 2.2.  Most organic
and inorganic compounds commonly found at trace levels in the atmosphere do not absorb strongly
in this region (Gordon et al., 2017; Kochanov et al., 2019). For these compounds to interfere with
the CRDS measurement, their mixing ratios would need to be very high (>10s of parts per million



by volume, ppmv). Under conditions where the peak shape is compromised by the presence of
interfering absorbing species or instrument instabilities (e.g. cavity pressure fluctuations), the
instrument fitting is interrupted and the "bad" data is flagged, thereby allowing simple quality
control (see Figure S2).
The limit of detection (LOD) of the CRDS analyzer is suitable for expected HCl levels in
the atmosphere. Instrument LODs were calculated as three times the Allan-Werle deviation (Figure
1) when overflowing a 15 cm inlet (3.17 mm i.d.) with zero air directed into the CRDS for ~10
hours. The LODs determined in the CRDS measurements for 2 second, 30 second, and 1 hour
integration times were 95, 18, and 2 pptv, respectively. Similarly, precision was determined from
the Allan-Werle deviation in the blank over the same 10 hours of zero air sampling. Precision in a
2 second, 30 second, and 1 hour integration time was 32, 6, and 0.8 pptv, respectively.

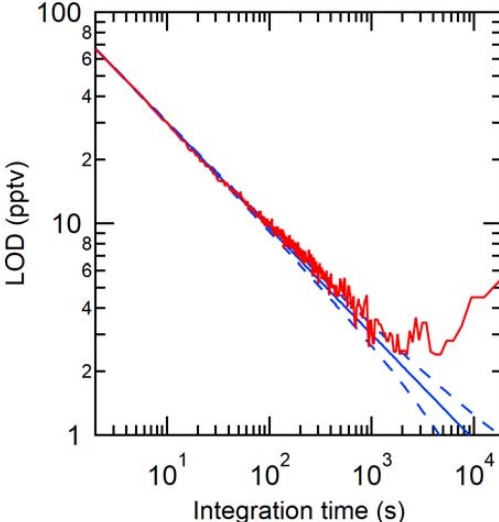


**Figure 1.** Allan-Werle deviation (3σ) in the optical cavity purged with zero-air (red line) shown with the
ideal deviation (no drift, solid blue line) and associated error in the deviation (dashed blue line).


### 3.2 Instrument performance

The CRDS has many advantages compared to methods previously used to measure HCl in the ambient atmosphere (Table 1). The LOD and precision of the instrument is comparable to prior high time-resolution methods, allowing changes in HCl mixing ratio of a few pptv to be measured. The accuracy/uncertainty is the hardest to compare due to the differences in assessment. A particular challenge is that other methods require external calibrations to determine accuracy and a stable, accurately calibrated HCl source at atmospherically relevant mixing ratios is challenging to obtain (Lao et al., 2020; MacInnis et al., 2016). In contrast, spectroscopic techniques offer a distinct advantage as they are absolute measurements and accuracy determinations rely on propagating uncertainty in the measured parameters (i.e. wavelength and the time constant $\tau$). In the absence of determining accuracy of the CRDS from its operating parameters we use the deviations in our intercomparisons to estimate the accuracy of the full system, i.e. the instrument and ambient sampling inlet combined. We measured that the accuracy of the analyzer ranges from 5 to 15%. This is a conservative range based on the methods we used to validate the instrument, which is further described in Section 3.3. The response time of the CRDS used in this work is fast compared to most measurements; the limitation for all online-line methods compared in Table 1 is not the measurement frequency, but rather the time required for HCl to adsorb and desorb from the inlet to the sample stream, which is discussed further in Section 3.5. Lastly, instrument size and power consumption of this CRDS are much lower than many other techniques and are major advantages when considering use in the field, particularly for mobile platforms.



**Table 1.** Performance characteristics of CRDS HCl analyzer compared to previously reported
methods.

| Instrument | LOD | Accuracy/ Uncertainty | Precision | Measurement frequency | Instrument Size | Power Consumption | Reference |
|---|---|---|---|---|---|---|---|
| *Near-IR CRDS* | <18 pptv[a] (30 sec) | 5–15% | 6 pptv (30 sec) | 2 s[e] | 31.75 kg 43.2 x 17.9 x 44.6 cm | 110 W (analyzer) 75 W (pump) | This study |
| *Near-IR CRDS* | 60 pptv[a] (1 min) | >10% | 20 pptv (1 min) | <15 s | NR | NR | (Hagen et al., 2014) |
| *Aircraft laser infrared absorption spectrometer* | 33 pptv[b] | 10 % | 0.1 ppbv (30 sec) | <30 s | 72 kg | NR | (Voss et al., 2001; Webster et al., 1994) |
| *Quartz-enhanced photoacoustic spectroscopy (QEPAS)* | 550 ppbv[a] | NR | 526 ppbv | NR | NR | NR | (Ma et al., 2016) |
| *Acetate CI-ToF-MS* | 97 pptv[a] | 30% | 32.3 pptv | <1 s | 59 x 42 x 83 cm | <2000 W peak | (Crisp et al., 2014) |
| *Iodide CI-HR-ToF-MS* | 30 pptv[a] (30 sec) | 30% | 53.3 pptv | 0.22 s | ~ 59 x 42 x 83 cm | <2000 W peak | (Lee et al., 2018) |
| *Sulfur pentaflouride ion trap CIMS* | 66 pptv[a] | 10% | 22 pptv | 1.6 s | NR | NR | (Jurkat et al., 2010) |
| *APCI-MS-MS* | 335 pptv[a] | NR | NR | 5 s | NR | < 17.5 kW | (Karellas et al., 2003) |
| *Tandem mist chamber and IC-CD* | 48 pptv[c] | >25 % | 24 pptv[c] | 2 h | NR | NR | (Keene et al., 2007, 2009) |
| *Annular denuder and IC-CD* | 6.9–42 pptv[d] | 10% | NR | Hours–Days | >10 kg | 400 W (sampling equipment only) | This study |

a: 3σ, b: predicted assuming a minimum detectable line-center absorption of $1 \times 10^{-5}$ can be achieved in 30 s, c:
precision (σ) determined from averaged paired measurements in 2 h samples on IC-CD and LOD was calculated at





2σ, d: 3σ calculated range for a 24-hour sampling time from three denuder method blanks, e: instrument data reporting
frequency. The true measurement frequency will also be affected by surface effects, as described in section 3.4, and
NR: not reported.
**3.3 Laboratory and ambient intercomparison**

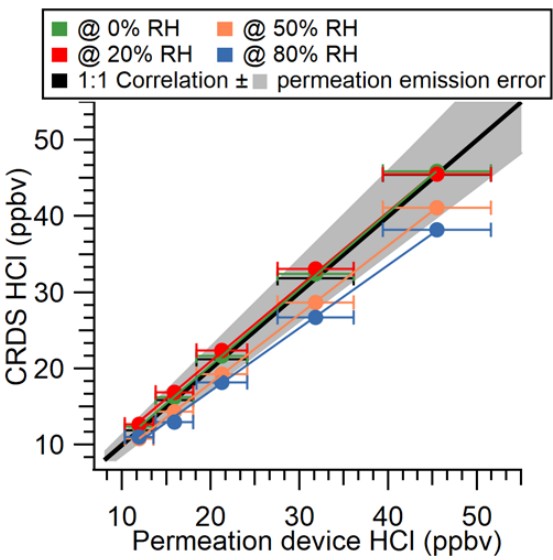


**Figure 2.** Comparison of CRDS HCl measurements and output from an HCl permeation device over a
range of RHs. Error bars in the x direction represent propagated IC measurement error, while error bars in
the y direction represent the standard deviation of online sampling plateau for each mixing ratio (low
magnitudes mean these error bars do not extend beyond the points). A 1:1 correlation (solid black line) is
shown with the uncertainty in the permeation device emission rate (shaded grey area).

275          We compared the CRDS analyzer-measured HCl with the gas standard mixing ratios

provided by an IC-certified PD under dry conditions and observed a close to 1:1 correlation (Figure
2). When the calibration gas entrained in flows of higher RH (≥ 50 %) a negative bias was
observed, although the measurements generally remained within the quantified error in the PD
output. Negative bias from the provided HCl mixing ratios at 50 % and 80 % RH were 9.6 % and
14.9 %, respectively. As described above, there is no spectroscopic water absorption interference
in the HCl measurement indicating that water increased HCl losses to gas handling surfaces for





experiments conducted in humidified air. Inlet surface effects are well established for gaseous
strong acids and bases, as these compounds readily sorb at interfaces (e.g. Eisele and Tanner
(1993), Kim et al. (2008), Neuman et al. (1999), Pszenny et al. (1993), Roscioli et al. (2016)). The
comparison presented here is a best-case scenario because the sampled mixing ratios were much
greater than expected in the ambient atmosphere, and therefore less likely to be impacted by
surface effects. Surface effects under humid conditions necessitated the mitigation and
quantification efforts described further in Section 3.5. To practically validate the CRDS under real-
world conditions, an ambient intercomparison was performed over a period of 7 days (4–11 April

290    2019).

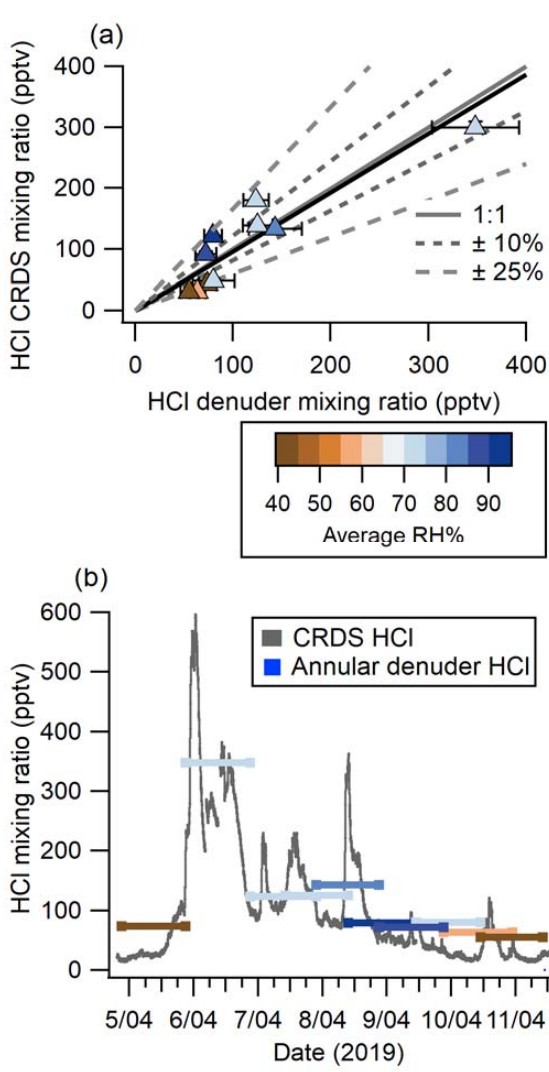


**Figure 3.** (a) Comparison of HCl measured 4–11 April 2019 using annular denuders and CRDS (averaged
to the collection time of denuders). Denuder error bars are derived from the error in the IC calibration,
standard deviation of method blanks, and extraction recovery. CRDS measurement errors are the precision
in a single measurement combined with data loss for flagged instrument errors. Also shown are a 1:1
correlation line (solid grey), 10 % (short grey dash) and 25 % (long grey dash) deviation from 1:1, and the
orthogonal distance regression (solid black). Points are coloured by the average RH during sampling. (b)
Continuous HCl mixing ratio timeseries by CRDS overlaid with averaged 24-hour denuder measurement
analyzed by IC with lines coloured according to the average RH during sampling.


Online HCl detection by CRDS showed good agreement with HCl mixing ratios quantified
from ten annular denuder extracts collected according to EPA Compendium method IO-4.2
(United States Environmental Protection Agency, 1999), which is a standard offline method for
quantitation of acidic atmospheric gases (Figure 2a). Measurements from the 2 instruments were
linearly related with a slope of 0.97 ± 0.15, as determined by orthogonal least distance regression,
with a y-intercept of -0.001 ± 0.021. Half of the measurements are within 10 % of a 1:1 correlation
and the remaining half fall within 25 %. To further validate the comparison a linear correlation
coefficient (see Figure SI4) of 0.93 ± 0.14 was determined for the two methods and shows good
agreement with the orthogonal least distance regression.  Changes in RH had no systematic bias
on the correlation. Our intercomparison indicates that CRDS measures HCl with comparable
results to those obtained by carbonate-coated annular denuders. While the latter requires offline
analysis, the CRDS has the additional benefit of continuous high time resolution measurements at
0.5 Hz and dramatically better precision.
Although average HCl measurements between the CRDS and denuders agreed well, much
of the useful temporal variability were lost in the time-integrated denuder data (Figure 2b). For
example, from 19:00 April 5 to 01:00 April 6 the CRDS measured mixing ratios between 91 and
598 pptv. This rapid change of mixing ratios is not captured by the 24-hour average denuder-
measured mixing ratio of 348 pptv. The fast time response of the CRDS also captured other rapidly
changing HCl features such as the peak observed on between 00:00–06:00 on April 7. The 6-hour
event started at 80 pptv and increased at a rate of 1.2 pptv min$^{-1}$ to 230 pptv over ~120 minutes,
followed by a decrease at a rate of 0.5 pptv min$^{-1}$ for ~240 minutes to 98 pptv. The fast time
response of the CRDS on the order of minutes is crucial when applying the technique to the real
atmosphere for the purpose of fully constraining the sources and sinks for HCl, for which many
precursors have similar lifetimes, and ultimately improve our understanding of the Cl budget
(Crisp et al., 2014).
Results from the laboratory and ambient intercomparisons were used to determine the
accuracy of the HCl analyzer as 5 to 15 %. The lower bound of uncertainty (5 %) was determined
from the laboratory intercomparison under the optimal dry conditions (Figure 1). The upper bound
of uncertainty (15 %) was consistent across the laboratory intercomparison under the highest RH
(80%) conditions tested (Figure 1) and the standard deviation of the orthogonal distance regression
slope from the ambient intercomparison (Figure 2a).
**3.4 Sampling line and instrument response time assessment**
We have thus far demonstrated the efficacy of the CRDS for accurately analyzing gas
standards and ambient HCl. However, the potential for sampling losses or desorption sources of
surface-active gases that could affect the quality of such measurements is ubiquitous, and the study
of these effects are well established (Crisp et al., 2014; Ellis et al., 2010; Pollack et al., 2019;
Roscioli et al., 2016). This makes quantification a challenge as there are typically long
equilibration times associated with signal stabilization. Long equilibrations make fast time
response detection difficult without first characterizing line sorption and desorption, followed by
making inlet modifications to minimize losses (Deming et al., 2019; Ellis et al., 2010; Pagonis et
al., 2017). To ensure accurate field measurements of HCl, a characterization of the magnitude of
HCl loss and desorption during sampling was made. The response time of an instrument to a rapid
change in HCl can be calculated by both the time it takes for the measurement to go from zero to
100 % of the HCl quantity being delivered, as well as the time it takes to return from the HCl
quantity being delivered back to zero.

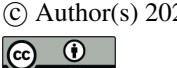

Inlet and instrument surface effects for surface active gases such as HCl can be
characterized by fitting decay curves to a double exponential (Ellis et al., 2010; Moravek et al.,
2019; Pollack et al., 2019; Zahniser et al., 1995);
$$y = y_0 + A_1 e^{\left(-\frac{t-t_0}{\tau_1}\right)} + A_2 e^{\left(-\frac{t-t_0}{\tau_2}\right)} \quad E2$$


Where y is the mixing ratio of HCl, $y_0$ is the mixing ratio at the end of the decay, $A_1$ and $A_2$ are
proportionality coefficients that determine how much the decay is governed by $\tau_1$ and $\tau_2$
respectively, t is the time elapsed, and $t_0$ is the initial time. The first time constant ($\tau_1$) represents
air exchange within the instrument, while the second ($\tau_2$) is the surface interaction equilibrium
time between HCl adsorbed to surfaces and the overlying airstream mixing ratio. The only term in
this equation that can be optimized for the CRDS is $\tau_2$, which can be reduced by decreasing the
amount of time HCl interacts with inlet surfaces. The sampling flow rate and cavity temperatures
are constant for the commercial software and not adjustable, fixing the value of $\tau_1$. In most reports
$\tau_1$ represents the largest change in signal and represents instrument response time. For example,
Whitehead et al. (2008) found that $\tau_1$ values governed >75% (i.e. $A_1$) of changes in measured $NH_3$.
For the CRDS, measured values of $\tau_1$ were between 5 and 10 seconds under all conditions and
could be improved with a faster inlet flowrate for the CRDS to subsample from.
An additional set of experiments were undertaken in which a 24 ppbv HCl standard was
sampled while varying RH from 0 to 33% (Figure 4). The effect of RH on the response time of the
CRDS was measured using a method similar to that described in Section 2.4. The HCl standard
gas was sampled over three 10 min pulses at each RH (± 2%). The HCl standard was introduced
into a 3.17 mm i.d. PFA tube and 10 cm long inlet line. Data was background corrected to levels
measured prior to the standard addition calibration and the signal was normalized to the HCl



enhancement during the final 10 seconds of each calibration pulse. Due to a lack of inlet
characterization for systems measuring HCl, we compare our decay constants to literature values
for compounds with similar surface-active properties (e.g. $HNO_3$ and $NH_3$). The instrument
exchange rate ($\tau_1$) values for spectroscopic methods measuring $HNO_3$ and $NH_3$ are generally faster
than our measured values for HCl. However, it should be noted that measurements of $\tau_1$ reflect
differences in sampling flow rates and internal volume and are likely affected by the internal filters
present in the HCl CRDS. Typically, $\tau_1$ was <2 seconds (Ellis et al., 2010; Pollack et al., 2019;
Roscioli et al., 2016), but could be as high as 4.5 seconds for larger pulses (1 ppmv) of analyte
(Roscioli et al., 2016). We observed a highly variable surface interaction equilibrium time constant
($\tau_2$), with values ranging between 97 and 350 seconds. Reported values of $\tau_2$ for other surface-
active gases are similarly variable, with values <50 seconds for heated short clean inlets (Ellis et
al., 2010; Pollack et al., 2019; Roscioli et al., 2016), and ~300 seconds for a contaminated inlet
measuring $NH_3$ (Pollack et al., 2019). Major differences in the surface area between our instrument
and the instruments we compare here are likely to cause $\tau_2$ discrepancies. Our method employs the
use of two HEPA filters that increase the gas to surface interactions by a greater degree.

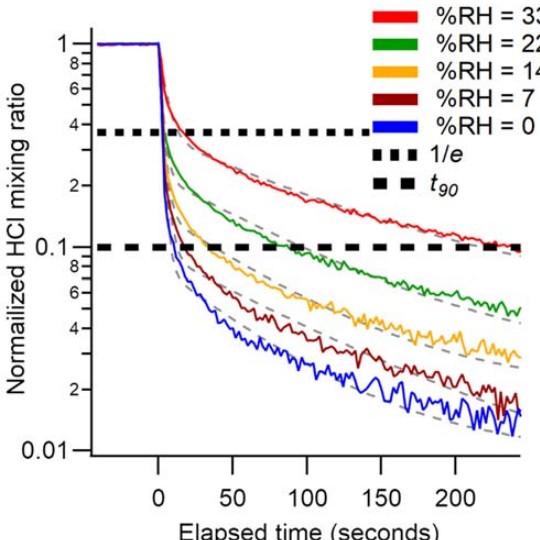

**Figure 4.** Background corrected and normalized signal decay curves observed for pulsed HCl (24 ppbv) performed at various RHs. Dashed grey lines represent a double exponential fit to the average of three cycles at each water mixing ratio. The short and long dashed black lines indicate 37 % (1/e) and 90 % ($t_{90}$) decrease from the initial signal, respectively.

Another method for quantifying response time is by calculating the e-folding (1/e) signal loss with respect to time. Calculated e-folding response times demonstrated the fast exchange within the system with values comparable to $\tau_1$. Similarly, a signal decrease of 90% ($t_{90}$) illustrates the total decay of the sampling line. Using e-folding response time and $t_{90}$ offers a better visual understanding of the relative roles for instrument and inlet responses and the impacts of increasing RH. We summarize the double exponential decay constants, e-folding response time, and $t_{90}$ for the rise and decay of HCl mixing ratios from pulses delivered to the instrument in Table 2 and SI tables 1–4. Increasing the RH increases the response time of the inlet (Figure 4). At the highest experimental RH (33 %), $\tau_2$ is increased by almost a factor of two compared to dry conditions, from seconds to minutes. At lower mixing ratios the higher RH increased the response times to a greater extent (Table 2 and Tables S1–S4). The increased response time at high ambient RH would



not compromise stationary measurements in which HCl mixing ratios changed on time scales of
minutes to hours but would not capture more rapid changes. The largest impact on CRDS time
response likely comes from unavoidable effects of partitioning on the large surface area of the
HEPA filters located before the optical cavity. Minimizing inlet effects for the CRDS where
observation of HCl mixing ratio changes over <1 min is required (e.g. aircraft or mobile
measurements) are most important. The wall interactions of HCl can be reduced by increasing the
inlet flow rate and/or decreasing the tubing length (Pagonis et al., 2017).

408       Physical approaches to improving inlet response include inlet material substitution,

heating, and pressure reduction, to reduce adsorption of surface-active analytes through removal
of surface water and promoted mass transfer of analytes to the gas phase (Sintermann et al., 2011).
Deming et. al. (2019) found that PFA tubing had the lowest delay times for semivolatile
compounds and would likely extend to small polar molecules like HCl. For a clean thermally-
equilibrated inlet, HCl artifacts can be minimized, but if semivolatile aerosol chloride is sampled
(e.g. $NH_4Cl$), the thermodynamic equilibrium can be shifted to result in a positive bias in the HCl
measurement, equivalent to similar considerations when measuring $HNO_3$ and $NH_3$ (Ellis et al.,
2010; Sintermann et al., 2011; Whitehead et al., 2008). While HEPA filters prevent aerosol from
entering the cavity, their elevated temperature (45 °C) could lead to volatilization bias and
therefore the use of an inlet filter held at ambient temperature to reduce such effects is
recommended at a minimum.

420       Chemical approaches can also help mitigate adsorption of surface-active molecules to inlet

surfaces through derivatization or passivation. The silanization of glass to form an inert fluorinated
or silicon coating on a virtual impactor or the introduction of a gaseous fluorinated compound that
adsorbs competitively to instrument surfaces in place of the analytes have been demonstrated to



substantially reduce surface adsorption on PFA (Ellis et al., 2010; Moravek et al., 2019; Pollack
et al., 2019; Roscioli et al., 2016; Wilkerson et al., 2021). However, environmental impacts must
be considered when constantly adding fluorinated compounds to sampling flows as they may have
deleterious environmental consequences (Cousins et al., 2020). In particular,
perfluorobutanesulfonic acid, the fluorinated chemical suggested for passivating inlets (Roscioli
et al., 2016), is subject to usage restrictions in some European countries (ECHA, 2020) based on
potential negative human and environmental health impacts (Benskin et al., 2012; Sunderland et
al., 2019). The high surface activity of perfluorobutanesulfonic acid is likely to cause issues in the
analyzer used in this work because the gas sample comes into direct contact with the high
reflectivity mirrors. Acid deposition onto mirrors will degrade their reflectivity.
**Table 2.** Summary of the fit parameters for the double exponential decay curves as a
function of mixing ratio and humidity, e-folding response time, and $t_{90}$.

| Mixing ratio (ppbv) | Residence time (seconds) | RH (%) | $\tau_1$ (seconds) | $\tau_2$ (seconds) | $1/e$ (seconds) | $t_{90}$ (seconds) |
|---|---|---|---|---|---|---|
| **12** | 0.021 | 0 | 10.3 | 123 | 26.5 | 101 |
| **16** | 0.028 | 0 | 9.6 | 200 | 24.7 | 82 |
| **21** | 0.037 | 0 | 10.1 | 300 | 26.4 | 74 |
| **24** | – | 0 | 2.7 | 97 | 2.5 | 10 |
| | | 7 | 5.0 | 124 | 2.5 | 18 |
| | | 14 | 5.0 | 114 | 2.5 | 32 |
| | | 22 | 5.0 | 123 | 3.9 | 86 |
| | | 33 | 10.0 | 189 | 16.1 | 239 |
| **32** | 0.056 | 0 | 9.4 | 188 | 24.5 | 62 |
| **45** | 0.079 | 0 | 9.7 | 350 | 25.6 | 54 |



## 4. Conclusions

The suitability of a CRDS analyzer for measuring ambient atmospheric HCl were explored
through calibration, inlet and analyzer sampling challenges, and intercomparison to established





atmospheric sampling techniques for strong acids. In comparison to other reported
instrumentation, the CRDS is shown performing similar or better than the most sensitive HCl
measurements reported. As with many in situ measurements of HCl, the most significant limitation
is adsorption/desorption loss and release on inlet surfaces, with the deposition effects increasing
with increasing RH and decreasing HCl mixing ratios. Given the longstanding knowledge of these
issues for surface active gases, such as $HNO_3$ and $NH_3$, there are a variety of chemical and physical
options, discussed in this study, to mitigate inlet effects and achieve faster response times for the
CRDS. Increasing the flowrate of the sampling inlet, while maintaining laminar flow, is the
simplest approach to reducing surface effects discussed in Section 3.4. Spectra capturing errors in
the measurement of HCl for the CRDS can occur at high levels of VOCs (e.g. near emission point
sources or biomass burning plumes) or instrument instabilities (e.g. pressure fluctuations),
however potential instrument errors are minimal under most operating and atmospheric conditions.
Finally, comparison with annular denuder observations agreed within the combined uncertainties,
with the CRDS measurement rate demonstrating the power of capturing transient events that are
important to constraining atmospheric chlorine chemistry (e.g. photolysis of precursors,
thermodynamic partitioning, and direct emissions).

## Author Contributions

TCF, PRV, and KERD collected and analyzed the data. TCF, PRV, JAN, SSB, TCV, and CJY
conceived of and designed the experiments. Funding was obtained by CJY. The manuscript was
written by TCF with input from all authors.

## Competing Interests

The authors declare that they have no conflict of interest.





## Acknowledgements

We acknowledge the Natural Sciences Engineering and Research Council of Canada and York University for funding. We thank Andrea Angelucci and Sonya Daljeet for assistance with data collection.

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
