# Peer review of "Validation of a new cavity ring-down spectrometer"

_Atmospheric Measurement Techniques, 2021_

## Author Comment (AC1)

**Reviewer 1**

We thank the Reviewer for their constructive comments on our manuscript. Please find our responses below highlighted in yellow, and changes to the manuscript in **bold**.

"The manuscript entitled "Validation of a new cavity ring-down spectrometer for measuring tropospheric gaseous hydrogen chloride" by Furlani et al. discusses the implementation of a commercial spectroscopic HCl instrument for ambient air measurements. The study indicates that the Picarro HCl CRDS instrument can be implemented with low detection limits, high precision, high time resolution, and similar/better accuracy compared to a cadre of other trace HCl measurement techniques. Discussion of sampling and analytical issues is thorough and thoughtful. Statistical treatment and reporting is adequate. Studies of inlet effects is detailed and provides important information for potential users in the future."

This paper is high quality and should be published in *Atmospheric Measurement Techniques* pending consideration of a small number of comments provided below.

Line 229 and Figure 1: Please provide a reference (within the associated text) to the approach of using the Allan-Werle deviation.

Response: Added reference to Hagen et. al. "Instrument LODs were calculated as three times the Allan-Werle deviation (Figure 1**, Hagen et al., 2014**) when overflowing a 15 cm inlet (3.17 mm i.d.) with zero air directed into the CRDS for ~10 hours." (P11, L239)

Lines 246-248: "spectroscopic techniques offer a distinct advantage as they are absolute measurements and accuracy determinations rely on propagating uncertainty in measured parameters" This statement is then met with the reality of the present study [in subsequent lines of this paragraph] in which concurrent denuder measurements are used to determine accuracy. Why not stand upon the 'distinct advantage' that spectroscopic measurements provide? [See also: next comment.]

Response: We take the Reviewer's point. We have clarified our text to explain our reasoning that while the spectroscopic detector is itself absolute, it does not represent the full method. Our validations test the full method, including sample collection. The following text has been added:

**"We assess the total method uncertainty using intercomparisons with the gold standard for atmospheric acid detection (EPA Compendium method IO-4.2, United States Environmental Protection Agency** (United States Environmental Protection Agency, 1999)**) due to the greater uncertainty when considering the potential total system error from sorption/desorption to all sampling surfaces (i.e. instrument and inlet)."** (P12-13, L261-265)

Lines 248-251: "In the absence of determining accuracy of the CRDS system from its operating parameters…" Is this due to lacking knowledge of these parameters due to the commercial nature of the instrument or are the intercomparison differences seen to be greater in importance, greater in magnitude, or both? Overall -- please clarify the motivation for the accuracy determination approach that was undertaken.

Response: The Reviewer is correct. We are unable to determine these parameters due to the commercial nature of the instrument. As described above, we use intercomparisons to validate the complete method, including any physical surface effects of the total method. We suspect very little of the total method error is related to the detector parameters.

Calibration questions: What is the impact of uncertainty in the permeation tube HCl concentration on the mixing ratio determination by the CRDS instrument? For instance, why aren't the error bars in the horizontal direction in Figure 2 comparable to those in the vertical direction? How can a measurement have lower uncertainty than the standard from which it was calibrated? Please clarify this issue. Perhaps I am confused about details of the study, but so too may be a future reader.

Response: The Reviewer raises a good point. The errors in the horizontal represent systematic errors propagated throughout the analysis of the collected samples and the vertical error bars represented the standard deviation of the CRDS signal once stabilized. One reason the error in the horizontal is much greater is the need for a calibration in the offline analysis of the IC samples. We use our validation experiments to determine an uncertainty in the CRDS, so we choose not to represent that error here.

To clarify, we have removed any instance in the text where we refer to the intercomparison experiments as calibrations due to the higher associated error in the offline extraction techniques used.

"We configured a commercial HCl cavity ring-down spectrometer (CRDS) for sampling HCl in the ambient atmosphere and developed  validation techniques to characterize the measurement uncertainties." (P1, L14-L16)

"The accuracy was determined to reside between 5–10%, calculated from laboratory  and  ambient air intercomparison**s** with annular denuders." (P1, L18-L19)

"The 140 ng min$^{-1}$ of HCl in dry $N_2$ from the PD was mixed into a zero air dilution flow of 2.1 to 8.0 L min$^{-1}$, to provide standard addition HCl **mixing ratios** that ranged from 12 to 45 ppbv." (P8, L169-L171)

"When the **HCl** gas entrained in flows of higher RH ($\geq$ 50 %) a negative bias was observed, although the measurements generally remained within the quantified error in the PD output." (P15, L292-L294)

"Data was background corrected to levels measured prior to the standard addition  and the signal was normalized to the HCl enhancement during the final 10 seconds of each **HCl** pulse." (P20-P21, L83-L385)

The suitability of a CRDS analyzer for measuring ambient atmospheric HCl **was** explored through **laboratory and ambient air intercomparisons, assessing their inlet and analyzer sampling challenges to established atmospheric sampling techniques for strong acids.** (P23-25, L462-464)

Figures 2 and onward: Numbering of figures in textual references is often (always?) incorrect.

Response: The numbering in references has been corrected throughout.

Line 362: Please clarify whether the effect of sampling line flowrate on $\tau_1$ was investigated experimentally or if this point is a supposition.

Response: This is a supposition from conclusions drawn in the literature for similar methods. The sampling flowrate for this instrument is fixed and therefore we could not directly test this. We have added the following to clarify.

"The sampling flow rate and cavity temperatures are constant for the commercial software and not adjustable, **therefore changing the value of $\tau_1$ was not explored in this study**." (P20, L372-L373).

Line 363: Change to "An additional set of experiments was…"

Response: Corrected.

**References**

Hagen, C. L., Lee, B. C., Franka, I. S., Rath, J. L., Vandenboer, T. C., Roberts, J. M., Brown, S. S. and Yalin, A. P.: Cavity ring-down spectroscopy sensor for detection of hydrogen chloride, Atmos Meas Tech, 7(2), 345–357, doi:10.5194/amt-7-345-2014, 2014.

United States Environmental Protection Agency: Compendium of Methods for the Determination of Inorganic Compounds in Ambient Air: Determination of reactive acidic and basic gases and strong acidity of atmospheric fine particles (<2.5 μm) (Compendium Method IO-4.2)., 1999.

---

## Author Comment (AC2)

**Reviewer 2**

We thank the Reviewer for their constructive comments on our manuscript. Please find our responses below highlight in yellow, and changes to the manuscript in **bold**.

"In this paper, the authors present a CRDS for HCl and evaluate its performance with an appropriate set of laboratory evaluations. They also report measurements of ambient HCl and compare these with another measurement technique; this comparison is done extremely well.

The findings are within the scope of AMT, substantive conclusions are reached about the instrument performance and field comparison, and the paper is well written. The instrument itself is not particularly novel since an extremely similar instrument was presented in AMT almost a decade ago (Hagen et al., 2014). This instrument does seem better than that one in some ways (mainly in the 30 s precision of 6 pptv, which is quite impressive), so I suppose publishing on this instrument is justified.

Still, there are some pieces that are missing or need revision. I discuss these in 'Broader Comments.' I'm calling these minor, but they do approach the threshold between 'minor' and 'major' revisions. I also have some decidedly minor suggestions, which follow as 'Specific Comments.' I recommend accepting for publication after these aspects are addressed."

Broader Comments:

**1)** The introduction should have a slighty more comprehensive discussion of HCl measurement techniques. There have been many cavity-enhanced HCl instruments developed in the past decade or so that are not mentioned here but should be. Los Gatos Research and Tiger Optics have already developed commercial, cavity-enhanced HCl spectrometers. Hagen et al., 2014 and Wilkerson et al., 2021 have also published on cavity-enhanced HCl instruments. Hagen's instrument is even a CRDS that measures HCl via its first overtone—the same as the instrument discussed here. So you should also explain how this instrument compares and contrasts with Hagen's since they are so similar.

Response: We thank the Reviewer for this suggestion to better represent spectroscopic techniques in the introduction and have added the following to the text.

**"Spectroscopic techniques offer distinct advantages over some previous methods. Spectroscopic techniques for measuring atmospheric HCl reported by Hagen et al.** (Hagen et al., 2014) **and Wilkerson et al. (2021) have shown the precedent for fast time response, as well as sensitive, selective, and robust detection. The portability and fast time response for instruments is of great importance for spatial resolution and is therefore a key factor for field deployment."** (P5, L90-L95)

**2)** There is no CRDS instrument schematic in this CRDS instrument paper. I understand that proprietary issues may limit how specific this can be, but please include at least a rough schematic of the instrument layout as either a main figure or supplemental figure.

Response: We have added a simple schematic of the CRDS to the supplement.

"The Picarro G2108 Hydrogen Chloride Gas Analyzer system was used for all analyses **(see Figure S1,** Dawe et al., 2019, www.picarro.com)." (P6, L119-120)

[Figure]

**Figure S1.** Simple schematic for the Picarro CRDS.

**3)** You state that tropospheric HCl mixing ratios vary between 10-1000 pptv. Yet, the data in Fig. 2 extends all the way up to 45 ppbv HCl and doesn't go below 10 ppbv HCl. You really need to be evaluating these effects closer to relevant, atmospheric levels of HCl. Ideally, you would dip below ppbv levels since those are the most relevant (as you further demonstrate in Fig. 3). Hagen et al., 2014 goes slightly below 1 ppbv HCl when they produced their correlation line (Fig. 8); they proposed an instrument extremely similar to this one and published that discussion in this very same journal (same with Wilkerson et al., 2021). There is, therefore, a clear precedent for a more rigorous evaluation than this. Having said that, I appreciate that laboratory assessments of sub-ppbv levels for HCl can get quite difficult. At bare minimum, you should remove the data above 30 ppbv and add at least two iterations of this experiment below 10 ppbv HCl.

Response: We appreciate the Reviewer's concern here. The levels we have generated are higher than typical atmospheric mixing ratios but validate the linearity and accuracy of the instrument in this range. As the Reviewer notes, it is challenging to generate an accurate HCl source at low mixing ratios. Thus, we have chosen to explore the lower HCl ranges using a field intercomparison. We have added text to clarify:

"We compared the CRDS analyzer-measured HCl with the gas standard mixing ratios provided by an IC-certified PD under dry conditions and observed a close to 1:1 correlation. **We explored 5 mixing ratios 12, 16, 21, 32, and 45 ppbv (Figure 2). These levels are higher than have been observed in the ambient atmosphere but demonstrate good signal response linearity.**" (P15, L287-L290)

"To practically validate the CRDS under real-world conditions **and atmospherically relevant mixing ratios**, an ambient intercomparison was performed over a period of 7 days (4–11 April 2019)." (P16, L302-L304)

Specific Comments:

Line 24: Recent reports of response times for HCl instruments are usually presented as 90% response times. Is that the case with this range? If not, please change this to a 90% response time range, so your instrument can be more easily compared with other recently reported ones (or at least explicitly call your current range a X% response time, where X is whatever this reported range corresponds to).

Response: We chose to use the more conservative $\tau_2$ estimate on our response time to reflect challenges with inlet effects. However, we do present the 90% response times in Table 2. We have also added the minimum 90 % response time to the abstract:

"**The minimum 90% response time was 10 seconds and the equilibrated** response time for the tested inlet was 2–6 minutes under the most and least optimal conditions, respectively." (P2, L24-L25)

Line 29: You should probably drop 'response time' from this list. First, a response time of 2-6 minutes is quite worse than recent HCl instruments discussed in scientific literature (Hagen et al., 2014; Wilkerson et al., 2021) and a bit worse than commercial HCl instruments (e.g. LGR's HCl analyzer is ~1 minute: http://www.lgrinc.com/documents/HCl_Datasheet2019.pdf). Second, you don't display response times in Table 1 where evidence of a favorable response time compared to other instruments would presumably be presented to the reader.

Response: It is difficult to compare response times for instruments that measure these high surface-active gases, as it often reflects the response of the whole system (i.e. instrument and inlet). Under optimal low humidity conditions, we can achieve a 90% response time of 10 seconds (see Table 2). Without in-depth knowledge of the inlets, flow rates, RH, etc., response times of different instruments cannot be directly compared. We have added text to clarify this point:

"Major differences in the surface area between our instrument and the instruments **to which** we compare are likely to cause $\tau_2$ **differences**. Our method employs the use of **three** filters that increase the gas to surface interactions**, and therefore increase our equilibrated response time $\tau_2$.**" (P21, L396-L399)

Line 49: Add just a couple sentences to explain where you would expect to see atmospheric HCl levels of 10 and 1000 pptv. For example, my understanding is that atmospheric HCl is elevated near oceans because sea salt spray provides a large source of chlorine to the atmosphere. (And if I'm wrong, then that's more evidence to support adding a couple sentences to briefly explain this to readers).

Response: The following text has been added:

"**Elevated levels of HCl are typically found near marine environments polluted with NO$_x$; where reactions involving the chloride in sea spray aerosols can be a major source of chlorine to the troposphere (Crisp et al., 2014; Finlayson-Pitts et al., 1989; Haskins et al., 2018; Wang et al., 2019).**" (P3, L50-L52)

Line 118: The first overtone has a very low intensity compared to the fundamental transition for HCl, and the fundamental transition is by far the most common transition used in HCl spectrometers (there are also ro-vibrational lines in the fundamental transition that are spectroscopically isolated from other atmospherically relevant species like H2O). You should clarify the language 'relatively high intensity' so the reader knows what this intensity is relative to.

Response: Text added for clarity:

"The first overtone (2-0 absorption band) of HCl is easily discernable from other absorbing species (e.g. H$_2$O, CH$_4$), has a relatively high intensity **compared to the fundamental absorption transition**, and is accessible to near-infrared (IR) diode laser light sources." (P6, L125-L129)

Line 234: The 30-second precision of 6 pptv is great—surprisingly great. Hagen et al. 2014 discussed a CRDS HCl instrument that also engaged a line in HCl's first overtone. This is pretty similar to your setup. Yet, their 60 s precision is 20 pptv. You should include a compelling explanation in this paper as to how your precision is so much better than any other HCl instrument, especially one that is so similar to yours.

Response: We agree that the performance of the commercial device is quite good. Unfortunately, we cannot compare the two instruments without detailed knowledge of the specific technologies. The commercial device is proprietary, and so we can only offer the references describing the methods as already presented in the Hagen et al. (2014) and Crosson (2008) references.

Line 234: Your response time is 2-6 minutes, but the precisions you report skip from 30 seconds to 1 hour. Can you provide a precision for an averaging time that is within range of or only slightly above your response time (e.g. 5 minutes or 10 minutes)?

Response: The Reviewer raises an excellent point. We have added precision for a 5-minute averaging time. *Also note that we corrected the reported value for the 2 second LOD and precision, which we noticed was mistakenly reported.

"The LODs determined in the CRDS measurements for 2 second, 30 second, **5 minute,** and 1 hour integration times were **66,** 18, **5,** and 2 pptv, respectively. Similarly, precision was determined from the Allan-Werle deviation in the blank over the same 10 hours of zero air sampling. Precision in a 2 second, 30 second, **5 minute,** and 1 hour integration time was **22,** 6, **2**, and 0.8 pptv, respectively." (P11, L240-L244)

Line 240: Change to, "**This** CRDS has many advantages compared to…" since you are comparing this instrument to another CRDS in Table 1.

Response: Changed as suggested.

Line 253: Again, this response time is not that fast compared to many recent HCl instruments (see earlier comment). Can you better contextualize this assertion?

Response: As described above, we cannot directly address this. A shared inlet intercomparison of HCl instruments would be beneficial to better understand the relative roles of inlets and detectors in determining instrument response times.

Line 261: Why does the Iodide Cl-HR-ToF-MS have a higher LOD (30 pptv) than precision (53.3 pptv)?

Response: We thank the Reviewer for pointing out this error. We have corrected the reported precision to match the LOD in the 30 sec integration.

| Iodide CI-HR-ToF-MS | 30 pptv[a] (30 sec) | 30% | 10 pptv (30 sec) | 0.22 s | ~ 59 x 42 x 83 cm | <2000 W peak | (Lee et al., 2018) |
|---|---|---|---|---|---|---|---|

Line 261: You allude to two recent cavity-enhanced HCl instruments in the main text that were published on in AMT (Hagen et al., 2014; Wilkerson et al., 2021), but only one is present in the table. There are also other commercial, cavity-enhanced instruments beside the one in this paper (LGR and Tiger Optics). If possible, can you add these missing ones to your table? (I say 'If possible' because their descriptions may be missing too much information to warrant inclusion).

Response: We have added data from Wilkerson et al. (2021) to the table. We have chosen to include only peer-reviewed data in Table 1. In our experience, specifications reported on data sheets are not held to the same rigour as peer-reviewed publications.

| Off-axis integrated cavity output spectrometer (OA-ICOS) | 78 pptv (30 sec) | <11% | 26 pptv (30 sec) | 1 s | NR | NR | (Wilkerson et al., 2021) |
|---|---|---|---|---|---|---|---|

Line 261: Hagen et al., 2014 has an accuracy <10%, not >10% (>10% reads as if the accuracy is at best 10% but could be infinitely worse). I believe that the Tandem mist chamber and IC-CD is the same situation (>25% when it should be <25%). Please correct these or just report the number without a '<' sign.

Response: We thank the Reviewer for noting this mistake. It has been corrected.

| | | | | | | | |
|---|---|---|---|---|---|---|---|
| *Near-IR CRDS* | 60 pptv[a] (1 min) | **<10%** | 20 pptv (1 min) | <15 s | NR | NR | (Hagen et al., 2014) |
| *Tandem mist chamber and IC-CD* | 48 pptv[c] | **<25 %** | 24 pptv[c] | 2 h | NR | NR | (Keene et al., 2007, 2009) |

Line 304: I think you mean to refer to Figure 3a.

Response: The numbering in references has been corrected throughout.

Line 425: Some of these references successfully used a silicon coating on their instrument, as you state on Line 422. Yet, your focus for the rest of this paragraph is on all the unpleasantness associated with fluorinated materials. What's your evaluation of silicon coatings, which lack all of these disadvantages? Please include that (even if it's just a few sentences).

Response: We thank the reviewer for their comment and have added the following text:

**"Silicon coatings on all plumbed surfaces have been successfully used for atmospheric HCl measurements (Wilkerson et al., 2021), and recommended for applications where PFA use is impractical. Although a direct comparison has not been conducted for HCl, PFA inlet material has been reported to yield better response times than silicon coatings for nitric acid (Neuman et al., 1999). Differences in instrument configurations and applications may warrant the use of different inlet materials and coatings for successful measurement of atmospheric HCl."** (P24, L449-L455)

**References**

Crisp, T. A., Lerner, B. M., Williams, E. J., Quinn, P. K., Bates, T. S. and Bertram, T. H.: Observations of gas phase hydrochloric acid in the polluted marine boundary layer, J Geophys Res, 119, 6897–6915, doi:10.1002/2013JD020992, 2014.

Crosson, E. R.: A cavity ring-down analyzer for measuring atmospheric levels of methane, carbon dioxide, and water vapor, Appl Phys B Lasers Opt, 92(3), 403–408, doi:10.1007/s00340-008-3135-y, 2008.

Dawe, K. E. R., Furlani, T. C., Kowal, S. F., Kahan, T. F., Vandenboer, T. C. and Young, C. J.: Formation and emission of hydrogen chloride in indoor air, , 29, 70–78, doi:10.1111/ina.12509, 2019.

Finlayson-Pitts, B. J., Ezell, M. J. and Pitts Jr., J. N.: Formation of chemically active chlorine compounds by reactions of atmospheric NaCl particles with gaseous N2O5 and ClONO2, Nature, 337(6204), 241–244, 1989.

Hagen, C. L., Lee, B. C., Franka, I. S., Rath, J. L., Vandenboer, T. C., Roberts, J. M., Brown, S.

S. and Yalin, A. P.: Cavity ring-down spectroscopy sensor for detection of hydrogen chloride, Atmos Meas Tech, 7(2), 345–357, doi:10.5194/amt-7-345-2014, 2014.

Haskins, J. D., Jaeglé, L., Shah, V., Lee, B. H., Lopez-Hilfiker, F. D., Campuzano-Jost, P., Schroder, J. C., Day, D. A., Guo, H., Sullivan, A. P., Weber, R., Dibb, J., Campos, T., Jimenez, J. L., Brown, S. S. and Thornton, J. A.: Wintertime gas-particle partitioning and speciation of inorganic chlorine in the lower troposphere over the northeast United States and coastal ocean, J Geophys Res Atmos, 123(22), 12,897-12,916, doi:10.1029/2018JD028786, 2018.

Keene, W. C., Stutz, J., Pszenny, A. A. P., Maben, J. R., Fischer, E. V., Smith, A. M., von Glasow, R., Pechtl, S., Sive, B. C. and Varner, R. K.: Inorganic chlorine and bromine in coastal New England air during summer, J Geophys Res Atmos, 112(10), 1–15, doi:10.1029/2006JD007689, 2007.

Keene, W. C., Long, M. S., Pszenny, A. A. P., Sander, R., Maben, J. R., Wall, A. J., O'Halloran, T. L., Kerkweg, A., Fischer, E. V and Schrems, O.: Latitudinal variation in the multiphase chemical processing of inorganic halogens and related species over the eastern North and South Atlantic Oceans, Atmos Chem Phys, 9(19), 7361–7385, doi:10.5194/acp-9-7361-2009, 2009.

Lee, B. H., Lopez-Hilfiker, F. D., Schroder, J. C., Campuzano-Jost, P., Jimenez, J. L., McDuffie, E. E., Fibiger, D. L., Veres, P. R., Brown, S. S., Campos, T. L., Weinheimer, A. J., Flocke, F. F., Norris, G., O'Mara, K., Green, J. R., Fiddler, M. N., Bililign, S., Shah, V., Jaeglé, L. and Thornton, J. A.: Airborne Observations of Reactive Inorganic Chlorine and Bromine Species in the Exhaust of Coal-Fired Power Plants, J Geophys Res Atmos, 123(19), 11,225-11,237, doi:10.1029/2018JD029284, 2018.

Neuman, J. A., Huey, L. G., Ryerson, T. B. and Fahey, D. W.: Study of Inlet Materials for Sampling Atmospheric Nitric Acid, Environ Sci Technol, 33(7), 1133–1136, doi:10.1021/es980767f, 1999.

Wang, X., Jacob, D. J., Eastham, S. D., Sulprizio, M. P., Zhu, L., Chen, Q., Alexander, B., Sherwen, T., Evans, M. J., Lee, B. H., Haskins, J. D., Lopez-Hilfiker, F. D., Thornton, J. A., Huey, G. L. and Liao, H.: The role of chlorine in global tropospheric chemistry, Atmos Chem Phys, 19(6), 3981–4003, doi:10.5194/acp-19-3981-2019, 2019.

Wilkerson, J., Sayres, D., Smith, J., Allen, N., Rivero, M., Greenberg, M., Martin, T. and Anderson, J.: In situ observations of stratospheric HCl using three-mirror integrated cavity output spectroscopy, Atmos Meas Tech, 14(5), 3597–3613, doi:10.5194/amt-14-3597-2021, 2021.